# Effect of Conservative Rehabilitation Interventions on Health-Related Quality of Life in Women with Upper Limb Lymphedema Secondary to Breast Cancer: A Systematic Review

**DOI:** 10.3390/healthcare11182568

**Published:** 2023-09-17

**Authors:** María Nieves Muñoz-Alcaraz, Antonio José Jiménez-Vílchez, Luis Ángel Pérula-de Torres, Jesús Serrano-Merino, Álvaro García-Bustillo, Rocío Pardo-Hernández, Jerónimo Javier González-Bernal, Josefa González-Santos

**Affiliations:** 1Interlevel Clinical Management Unit of Physical Medicine and Rehabilitation, Reina Sofía University Hospital, 14004 Córdoba, Spain; marian.munoz.sspa@juntadeandalucia.es; 2Córdoba and Guadalquivir Health District, Andalusia Health Service, 14011 Córdoba, Spain; jesus.serrano.merino.sspa@juntadeandalucia.es; 3Maimonoides Biomedical Research Institute of Córdoba (IMIBIC), Reina Sofía University Hospital, University of Córdoba, 14004 Córdoba, Spain; langel.perula.sspa@juntadeandalucia.es; 4Valle de los Pedroches Hospital, Andalusia Health Service, 14400 Pozoblanco, Spain; antonioj.jimenez.vilchez.sspa@juntadeandalucia.es; 5Faculty of Health Sciences, University of Burgos, 09001 Burgos, Spain; rph1001@alu.ubu.es (R.P.-H.); jejavier@ubu.es (J.J.G.-B.); mjgonzalez@ubu.es (J.G.-S.)

**Keywords:** breast cancer, lymphedema, quality of life, rehabilitation, conservative interventions

## Abstract

Breast cancer-related lymphedema (BCRL) of the upper limb is a very common condition in women undergoing breast cancer treatment; it can cause considerable alterations in the daily life of patients and a decrease in their health-related quality of life (HRQoL). Currently, there are many conservative therapies that try to palliate the symptoms, but the results are still controversial and there are still no globally accepted treatments. The purpose of this article is to determine the effect, according to the current available evidence, on HRQoL of different conservative interventions in the rehabilitation of BCRL in the upper limb in women. Eighteen articles that compared the effects of standard treatments, such as manual lymphatic drainage-based decongestive therapy or compression measures, and other newer treatments, including new technologies and other types of treatment programs, were reviewed. According to the results of this review, the most recommended modality for the improvement of HRQoL would be a complex decongestive technique without manual lymphatic drainage. Although there are clinical trials that have demonstrated the effectiveness of various treatments, the results of the positive effects on HRQoL remain highly controversial. There is a need to continue to develop studies to help guide therapeutic decisions that can promote HRQoL in women affected by upper limb BCRL.

## 1. Introduction

Breast cancer has become the most frequently diagnosed cancer worldwide, surpassing even lung cancer. According to the latest statistics, an estimated 2.3 million women were diagnosed in 2020, and the burden of breast cancer is projected to increase to more than 3 million new cases by 2040 [1,2]. The breast cancer incidence rates have increased by 0.5% annually during the four last decades, with 287,850 and 297,790 estimated new cases of invasive breast cancer diagnosed in the United States in 2022 and 2023 respectively [3,4]. Among the possible side effects of its treatments, such as surgery or radiation, is the appearance of lymphedema, a chronic inflammation caused by damage or overload in the lymphatic system, which cannot clear lymph quickly enough, causing it to accumulate. Incidence rates vary widely, but it is estimated that approximately 30% of women diagnosed with breast cancer develop breast cancer-related lymphedema (BCRL) [5]. Although lymphedema can appear in various areas, such as the breast, armpit, chest, or back, it most commonly appears in the upper extremity on the affected side [6], presenting various symptoms such as swelling, pain, discomfort, tightness, hypersensitivity, or lack of sensation. In this way, dermatological alterations, sleep disorders, limitations in performing activities of daily living, dissatisfaction with body image, interference with social function, and even anxiety and depression can appear, considerably reducing the health-related quality of life (HRQoL) of patients with BCRL [7,8,9].

Therefore, treatment is aimed at improving the functionality and HRQoL of patients. To date, the most commonly used conservative treatment is combined physical therapy that includes skin care, manual lymphatic drainage (MLD), the use of compression garments, and various recovery exercises [10,11]. However, although there are guidelines for the application of this type of therapy, there is still not enough solid data to recommend its use, so there is no globally accepted treatment or criteria to guide therapeutic decisions.

There are many studies and clinical trials that evaluate the effect of various conservative treatments on BCRL [12]. Many of them focus their analyses on objective measurements, such as the volume of the affected arm, and some also evaluate subjective measurements, such as the level of pain reported by patients. However, although these measurements are important and many studies have obtained positive results, sometimes adherence to these therapies is not optimal, as a consequence of irritation and other discomforts derived from the treatments themselves, the difficulty or time spent in their application, or their interference in daily activities or in certain social situations [13]. Therefore, it is evident that the best outcome for assessing the success of a treatment should be the HRQoL reported by the patients. Although the impact of BCRL on HRQoL impairment is widely cited in the literature, this has not been sufficiently investigated over the long term and the HRQoL domains most affected have not been identified [14]. Similarly, the effect of the different treatments available for lymphedema on HRQoL is an emerging field of study that is made difficult by the heterogeneity of assessment instruments chosen by different investigators to assess the clinical efficacy of the different therapeutic modalities used in BCRL on the different dimensions of HRQoL. In order to be able to directly compare the results of the different treatments, studies should be carried out using a single well-developed and validated common instrument to assess patients’ self-reported results as they will be decisive for therapeutic decision [15]; they also help the therapist to evaluate the effectiveness of the treatment; so far, it is not known which HRQoL questionnaire has the best psychometric policies regarding BCRL [16].

Therefore, this article aims to understand the effect of different conservative interventions on HRQoL during the rehabilitation of BCRL in the upper limb in women based on the currently available evidence.

## 2. Materials and Methods

### 2.1. Design

This systematic review was conducted following the PRISMA guidelines [17], including the checklist.

### 2.2. Search Strategy

The search was performed using the Web of Science, PubMed, Cochrane, and OTseeker databases, searching the title, abstract, and keywords for the following MeSH terms: “Breast cancer”, “Lymphedema”, “Rehabilitation”, and “Quality of life”. Table 1 shows the detailed search strategy in each database.

### 2.3. Eligibility Criteria

To determine the eligibility criteria, the PICOS criteria were used to determine studies that would be included in the bibliographic review, as shown in Table 2. The identified articles were considered for review if they were controlled clinical trials, reported HRQoL as a primary or secondary outcome, and compared any conservative intervention with either no intervention or another conservative intervention in women with BCRL in the upper limb.

### 2.4. Study Selection Process

The eligibility of identified trial reports was reviewed independently by three authors (NMA, AGB, and RPH). Full texts were obtained if there was a need to do so. Eligibility was resolved by discussion and adjudicated by a majority of the authors.

### 2.5. Quality Assessment and Certainty of Evidence

The studies’ methodological quality and risk of bias were assessed using the PEDro scale [18], which evaluates the internal validity and the statistical information of the study, ranging from 0 (worst quality) to 10 (highest quality). Scores ranging from 0 to 3 have a lack of methodological quality, scores of 4 or 5 have acceptable methodology, scores from 6 to 8 have good methodology, and scores of 9 or 10 have excellent methodology. Controversial opinions about study quality were resolved by discussion and adjudicated by the majority of the authors (NMA, AGB and RPH).

### 2.6. Data Analysis

Two of the authors (AGB and RPH) extracted data on trial design, sample demographic and clinical characteristics, type of conservative intervention, and HRQoL efficacy outcomes.

### 2.7. Articles Selection

Figure 1 shows the flow diagram of the eligible articles with the inclusion and exclusion criteria and data extraction. The study selection filter was initially based on the information in the title, and subsequently based on the abstract and full text if the abstracts did not contain the necessary information. In the full-text screening phase, the articles were checked to see if they met the inclusion criteria for this review.

Clinical case reports, scientific letters, bibliography reviews, protocols, studies that analyzed other populations (not female) or etiologies, studies that did not answer the research question or were not related to the main objective of the review, and low-quality scientific reports were excluded.

### 2.8. Quality of the Articles Included

Table 3 shows the PEDro scores. Half of the articles included in this review (50.00%) scored 9 or 10, indicating excellent methodology. Nearly half of the remaining articles (44.44%) obtained a score of 6, 7, or 8, indicating good methodology. Finally, one of the articles included (5.56%) obtained a score of 5 points, which indicates an acceptable methodology. The quality of the studies was very high due to the rigorous selection process, in which only clinical trials with adequate methodologies were selected.

The item with the lowest score was number 5 (patient blinding) since only two of the included studies (11.11%) met this criterion. This may be because due to the methodology of this type of trial, in which interventions are carried out directly with the patients; it can be difficult therefore for them to be blinded. On the other hand, half of the included studies (50.00%) met criteria 6 (therapist blinding) and 7 (rater blinding).

The rest of the criteria were met in all studies, with the exception of number 2 (random allocation) and 3 (concealed allocation); these were not met in one article (5.56%) because it was a single-arm study pilot clinical trial with a single group of patients in which they were their own controls. Another study (5.56%) did not meet criterion number 4 (group homogeneity) since the data from the groups were not similar at the beginning of the study.

## 3. Results

### 3.1. Sample Characteristics

Table 4 shows a summary of the main aspects of each of the articles included [19,20,21,22,23,24,25,26,27,28,29,30,31,32,33,34,35,36]. The eighteen trials included a total of 1293 patients, all women with stage I, II, or III BCRL in the upper limb, with an age range of 0–86 years. These clinical trials were conducted in the following countries: U.S.A. (n = 2), Scotland (n = 1), Israel (n = 1), Iran (n = 2), Taiwan (n = 1), Egypt (n = 1), Spain (n = 2), South Korea (n = 2), Australia (n = 3), Belgium (n = 1), and Canada (n = 2).

Most of the selected investigations were carried out using conservative techniques. All of them analyzed the impact of the intervention on HRQoL using different assessment tools.

### 3.2. Conservative Interventions

The most commonly used conservative treatments were compression elements, used in six of the studies (33.33%) either to verify their effect or as control therapy to evaluate other types of therapies. One of the studies analyzed the effect of different types of compression garment sizes, daytime only, daytime plus night bandage, and daytime plus nighttime garment. Although all treatments improved the HRQoL of patients, there was no difference between the three treatment options [32]. Another study conducted a similar study that compared a bandage compression system with a kinesiology tape system. There were hardly any differences, but in the long term, the bandage improved the domain of emotional function while the kinesiology tape worsened this domain. [24].

Manual drainage therapy is another common treatment. It was used in four of the included studies (22.22%). The results of this type of therapy are contradictory. Some studies reported benefits in HRQoL at an emotional level, reducing depression and improving sleep quality [21]; however, other studies did not obtain benefits or show differences with other types of usual treatments, such as compression measures [33], or with newer technologies, such as low-level laser therapy [19] or Flexitouch, which is a portable device that stimulates lymphatic drainage [20].

Other types of more advanced therapies, such as electrical moxibustion, did not obtain positive results [28], while others, such as low-intensity and low-frequency electrotherapy [27] or myofascial release [34], did improve the HRQoL of the patients.

Other programs also improved various aspects of patients’ HRQoL, such as aquatic lymphatic therapy [22], which favored emotional and social functions, an activity-oriented anti-edema proprioceptive treatment [29], which also favored the social dimension, or other programs, such as the Domiciliary Allied Health and Acute Care Rehabilitation Team (DAART) or the Strength Through Recreation Exercise Togetherness Care Health (STRETCH), which also contributed to increasing the level of quality of life reported by their participants [35]. Furthermore, another study showed that performing exercises with low and high loads and resistance can also improve general health and the perception of HRQoL [30].

Regarding care education, which was evaluated in two of the studies (11.11%), results showed that face-to-face education provided greater benefits than virtual education or education through social networks, improving the physical, emotional, and social components of the HRQoL [23,26]. In addition, another study assessed the effects of an educational and therapeutic program through the practice of yoga, obtaining benefits in the quality of life reported by the participants [36].

### 3.3. Health-Related Quality of Life Outcome Measures

The studies used various assessment tools and scales to assess the HRQoL of the patients included in the clinical trials. The most commonly used scales were the 36-Item Short Form Health Survey (SF-36) and the Functional Assessment of Cancer Therapy-Breast (FACT-B), which were each used in four studies (22.22%). In addition to these, the Upper Limb Lymphedema 27 Value (ULL-27), the EORTC Core Quality of Life questionnaire (EORTC QLQ-C30), and the EORTC Core Quality of Life questionnaire (EORTC QLQ-BR23) were also widely used in three items each (16.67%). Finally, although they were each only used once (5.56%), the Lymphedema Life Impact Scale (LLIS), McGill-QoL Questionnaire, Lymphedema Functioning, Disability, and Health Questionnaire (Lymph-ICF), and Lymphedema quality of life questionnaire (LYMQOL) were all used.

## 4. Discussion

Due to the increasing incidence of breast cancer worldwide, it is common for women to suffer from BCRL in the upper limb. This can mean a considerable change in several spheres of the daily life of the patients, even decreasing their HRQoL.

This article aimed to understand the effect of different conservative interventions during BCRL rehabilitation in women’s upper limbs based on the currently available evidence.

The results of this review conclude that the most recommended approach for the improvement of HRQoL in BCRL would be complex decongestive therapy (CDT), excluding the MLD component. Yan Lin et al. [37], in their systematic review and meta-analysis, also do not support the use of MLD to improve HRQoL. In the results of their systematic review, Belinda Thompson et al. [38] found that some studies reported positive effects of MLD on HRQoL, while other studies in their review reported no additional benefit of MLD as a component of a CDT. The results of the systematic review and meta-analysis by Shaimaa Shamoun et al. [39] recommended that patients perform CDT to improve their HRQoL, as do we with the findings of this review.

This systematic review also addresses the controversy about the impact of garment use on HRQoL in BCRL depending on whether the garment is worn during the day, night, or day and night; in one study, the intervention allowed the decision of whether to wear the garment during yoga was left to the participant’s discretion. Mona Al Onazi et al. [40] suggest that strict adherence to wearing the garment >12 h per day, while recommended, may not be necessary to achieve lymphedema control, that there are many barriers to use, and that use is varied. Sandy Hayes et al. [41] conclude in their systematic review that wearing compression garments during exercise provides no benefit or adverse effect, and Katarina Y. Blom et al. [42] state that some women may perceive practical and emotional problems with the compression garment, with a greater negative impact on their HRQoL than non-wearers.

The results of this research do not support the use of low-level laser therapy for the improvement of HRQoL. Dania Mahmood et al. [43] do not report this clinical application for use in BCRL either. 

Regarding the other treatment not recommended under the results of the present review, electrical moxibustion, Kyungsun Han et al. [28] also found no improvement in HRQoL after moxibustion intervention in their pilot study.

The researchers consider a limitation of their study to be the heterogeneity of the therapeutic modalities and outcome measurement instruments of the selected studies and not having included clinical trials that either compared the same intervention with the same assessment instrument or that assessed participants at the same stage of BCRL; they believe that these aspects should be addressed in future research.

Due to the scarcity of studies on lymphedema analyzing patient HRQoL, we did not limit the time period of the publications, with the articles’ publication dates ranging over 21 years in this review. However, it was found that conservative therapies, such as lymphatic drainage, are still being used and researched two decades later.

## 5. Conclusions

The results of this review confirm that there is limited and controversial information about the effects of the various conservative treatments for upper limb BCRL on HRQoL in women.

Manual lymphatic drainage: Of the 18 selected studies, 5 [19,21,27,31,33] compared the effect of MLD with other therapeutic modalities on HRQoL in people with BCRL. The MLD did not modify the HRQoL in BCRL, and its impact on HRQoL was equal to that of low intensity laser therapy, either used alone or in combination with it [19]. Combined with simple manual drainage (modified, as a self-help measure), whether the MLD was used first and followed by the simple one, or vice versa, it significantly improved the emotional, dyspnea, and sleep disturbance dimensions of the HRQoL in BCRL. However, simple lymphatic drainage did not produce improvements in HRQoL [21]. Compared with low-frequency and low-intensity electrotherapy, it did not produce improvements in HRQoL, either before or after electrotherapy. There were improvements with low-frequency and intensity electrotherapy for both general and specific dimensions of BC [27]. Adding MLD to other components of CDT did not provide additional clinically important benefits, but DMT did improve HRQoL, regardless of whether or not MLD was included [31]. When using MLD together with a compression bandage and a compression garment in an intervention group, compared to another group that only used compression garments, the MLD did not produce improvements on HRQoL, nor were there any differences between groups [33].

Decongestive therapy, standard or routine treatment: Of the 18 studies selected in this review, 5 of them [20,24,26,29,30] compared the effect of CDT with other therapeutic modalities on HRQoL in BCRL. Studying the effect of conventional CDT compared to CDT using an electronic device for MLD, CDT did not show any effect on HRQoL regardless of the modality and the order in which they were applied [20]. When comparing the use of two different bandages, components of the TDC, it was shown that significant improvements were achieved in the role-play dimension of the HRQoL in the immediate post-treatment group that used kinesiotaping, but in the follow-up at 3 months, the conventional taping group had significant differences in emotional function [24]. Contrasting its use in isolation, or in combination with face-to-face or virtual health education groups, it demonstrated a significant improvement in HRQoL in any of the three modalities, and this improvement was greater in the face-to-face education group [26]. Relating TDC with an experimental therapy that does not use compression, activity-oriented proprioceptive antiedema therapy (TAPA), showed improvements in HRQoL, but these were higher in the social dimension with the experimental treatment and were maintained three months after follow-up [29]. In their study, in comparison with low- or high-load resistance exercise interventions, TDC showed improvements in the physical dimension, but these were higher in the other two intervention groups that did not include it [30].

Lymphatic aquatic therapy: one of the articles included in this review [22] studied the effect of lymphatic aquatic therapy added to self-management measures compared to the use of self-management measures alone, finding that, when used in combination with lymphatic aquatic therapy, it improved the HRQoL in the emotional and social dimensions.

Virtual reality: two studies in this review evaluated the effect of the virtual context [23,25]. One of them [23] compared the effect of face-to-face and virtual self-care health education on the HRQoL and concluded that both modalities improve, but face-to-face to a greater extent, in the dimensions of global health, physical functioning, roles, emotional and social. The other [25] compared the use of games with Xbox with the performance of resistance exercises, finding that the use of games based on virtual reality obtained better results for HRQoL in terms of bodily pain, general health, and vitality.

Electronic moxibustion: The only study of those selected that used this therapeutic modality [28] found no significant differences in HRQoL in BCRL after the intervention.

Compression garments: The study that evaluated the effectiveness of the use of compression garments [32] showed that both daytime use, nighttime use, and daytime and nighttime use improved HRQoL to the same extent.

Myofascial release: One study [34] compared the use of myofascial release and TDC with placebo myofascial release and TDC, finding significant improvements in HRQoL before and after myofascial release.

Rehabilitation support services: Another of the selected studies demonstrated a greater short-term improvement in the physical, functional, and general dimensions of the HRQoL in patients who had received early home physiotherapy, psychosocial support, or exercise.

Yoga: In the comparison of the use of yoga and optional compression garment with respect to the usual self-care, the included study [36] concludes that there is a significant improvement in the HRQoL in BCRL in the group that used yoga in the short term, which was maintained at 8 and 12 months.

According to the results of this review, the most widely used therapeutic modalities to improve HRQoL in BCRL are MLD and CDT.

MLD does not appear to be a first-choice therapeutic option for HRQoL improvement in BCRL. Of the five studies that evaluated the use of the MLD, only 1 [21] of the 31 participants experienced significant improvements in HRQoL with its use, compared to the 379 participants from the other four studies that found no improvement. It is unknown what BCRL stage the patients were in when these improvements were obtained or if they were evaluated using a different instrument than the one from the other studies, i.e., the EORTC QLQ-C30. In fact, only two of the studies [19,27] shared an instrument, the FACT-B.

The TDC is an effective therapy for the improvement of the HRQoL in BCRL. Only one of the studies [20] found no significant improvements from its use in HRQoL when using the SF-36 as an instrument and for 10 participants in BCRL stages I, II or III, compared to a total of 237 participants from the other four studies that did experience short-term benefits in the role-playing [24] and physical [30] dimensions.

The HRQoL results of CDT could be enhanced if it is combined with face-to-face health education groups [26] or myofascial release [34].

Other recommended approaches for improvements in HRQoL are lymphatic aquatic therapy with self-management recommendations [22], face-to-face self-management group sessions [23], virtual reality-based Xbox games [25], low-frequency and intensity electrotherapy [27], activity-oriented proprioceptive anti-edema therapy (TAPA) [29], low- or high-load resistance exercise [30], rehabilitation support services [35], and yoga with optional garment use compression [36].

Regarding the use of compression garments, the results are controversial in this review, since it seems equally effective if they are used during the day, at night, or at night and during the day [26], or if use is optional, without appearing to alter results [30].

Low-intensity laser therapy [19] or electronic moxibustion [28] would not be recommended.

In light of the results of this review, the most recommended modality for improving HRQoL would be CDT without MLD.

However, there is a need to continue developing studies and clinical trials that help guide therapeutic decisions to promote HRQoL in women affected by BCRL of the upper limbs, using the same assessment instrument, specifically for the measurement of HRQoL in BCRL, for the same intervention.

Likewise, given the heterogeneity of approaches, many of them complex, it could be of interest to evaluate the impact of each of its components on the HRQoL in order to reach conclusive results.

It could also be convenient to standardize the maintenance of the effect of each of the approaches and the BCRL diagnostic criteria in the design of future research to determine which interventions are most effective in which stages and for how long.

## Figures and Tables

**Figure 1 healthcare-11-02568-f001:**
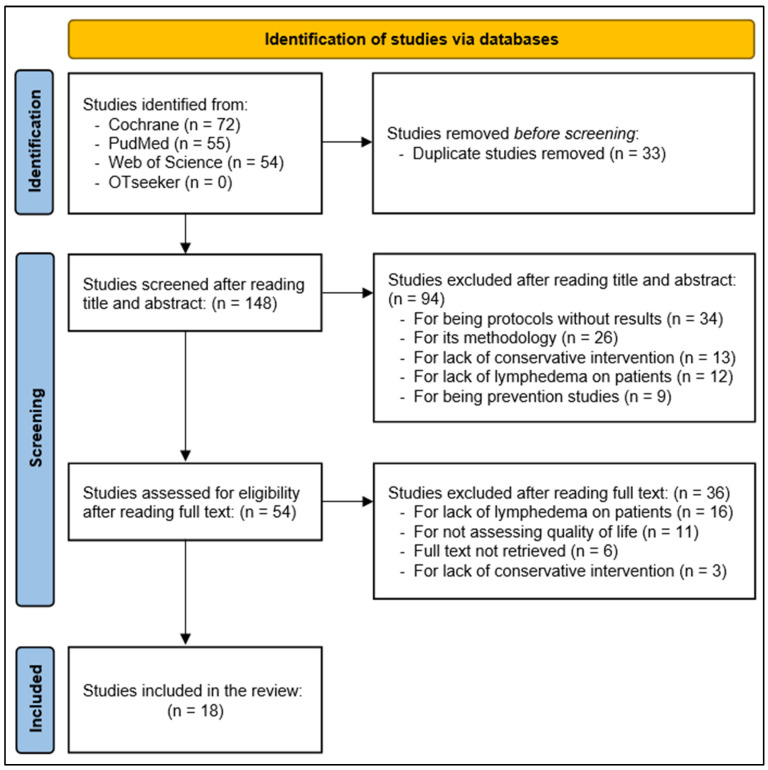
PRISMA Flow Diagram.

**Table 1 healthcare-11-02568-t001:** Database search strategy.

Database	Search Strategy
Cochrane	TITLE-ABS-KEY (breast cancer) AND TITLE-ABS-KEY (lymphedema) AND TITLE-ABS-KEY (rehabilitation) AND TITLE-ABS-KEY (quality of life)
PubMed	(((breast cancer [MeSH Terms]) AND (lymphedema [MeSH Terms])) AND (rehabilitation [MeSH Terms])) AND (quality of life [MeSH Terms]) Filters: Clinical Trial
Web of Science	(((TS = (breast cancer)) AND TS = (lymphedema)) AND TS = (rehabilitation)) AND TS = (quality of life)
OTseeker	[Title/Abstract] like ‘breast cancer’ AND [Title/Abstract] like ‘lymphedema’ AND [Title/Abstract] like ‘rehabilitation’ AND [Any Field] like ‘quality of life’

**Table 2 healthcare-11-02568-t002:** PICOS criteria for inclusion in the systematic review.

P	Population	Women with BCRL in the upper limb
I	Intervention	Any conservative intervention
C	Comparison	Conservative interventions with either no intervention or another conservative intervention
O	Outcomes	HRQoL
S	Study	Controlled clinical trials

**Table 3 healthcare-11-02568-t003:** Assessment of the articles using PEDro scale.

	Items											
	1 *	2	3	4	5	6	7	8	9	10	11	Total
Ridner SH et al. [19]	1	1	1	1	0	0	0	1	1	1	1	7
Wilburn O et al. [20]	1	1	1	1	0	1	1	1	1	1	1	9
Williams AF et al. [21]	1	1	1	1	0	0	0	1	1	1	1	7
Tidhar D et al. [22]	1	1	1	1	0	1	1	1	1	1	1	9
Haghighat et al. [23]	1	1	1	1	0	0	0	1	1	1	1	7
Tsai H-J et al. [24]	1	1	1	1	0	1	1	1	1	1	1	9
Basha MA et al. [25]	1	1	1	1	0	1	1	1	1	1	1	9
Omidi Z et al. [26]	1	1	1	1	0	1	1	1	1	1	1	9
Belmonte R et al. [27]	1	1	1	1	0	1	1	1	1	1	1	9
Han K et al. [28]	1	0	0	1	0	0	0	1	1	1	1	5
Muñoz-Alcaraz MN et al. [29]	1	1	1	1	0	1	1	1	1	1	1	9
Cormie P et al. [30]	1	1	1	1	0	0	0	1	1	1	1	7
De Vrieze T et al. [31]	1	1	1	1	1	1	1	1	1	1	1	10
McNeely ML et al. [32]	1	1	1	1	0	1	1	1	1	1	1	9
Dayes IS et al. [33]	1	1	1	1	0	0	0	1	1	1	1	7
Kim Y et al. [34]	1	1	1	1	1	0	0	1	1	1	1	8
Gordon LG et al. [35]	1	1	1	0	0	0	0	1	1	1	1	6
Loudon, A et al. [36]	1	1	1	1	0	0	0	1	1	1	1	7

Item 1: eligibility criteria; Item 2: random allocation; Item 3: concealed allocation; Item 4: group homogeneity; Item 5: patient blinding; Item 6; therapist blinding; Item 7: rater blinding; Item 8: key outcome collection; Item 9: intervention allocation; Item 10: between-group statistical comparisons; Item 11: key outcome measures report. * Item 1 influences external validity but not internal validity. This item is not used to calculate the PEDro score.

**Table 4 healthcare-11-02568-t004:** Characteristics of systematic review studies.

Authors (Year) City, Country	Study Design	Sample	Intervention/s	HRQoL Outcome Measure	Health-Related Quality of Life Results	PEDro Score
Ridner SH, Poage-Hooper E, Kanar C, Doersam JK, Bond SM, Dietrich MS (2013) [19] Nashville, U.S.A.	Pilot randomized clinical trial of three groups.	46 women older than 21 years, with stage I or II of BCRL in upper limb. Mean age = 66.6 ± 10.4	Group 1: Low-level laser therapy (n = 15). Group 2: Manual lymphatic drainage (n = 16). Group 3: Low-level laser therapy and manual lymphatic drainage (n = 15). Compression bandages were applied to all groups after treatment.	Upper Limb Lymphedema-27 (ULL-27). Functional Assessment of Cancer Therapy–Breast (FACT-B).	There were no significant differences between baseline and final HRQoL scores. There were no significant differences between the three groups.	7
Wilburn O, Wilburn P, Rockson SG (2006) [20] Stanford, U.S.A.	Prospective, randomized, and crossover study of two groups.	10 women with stage I, II or III of BCRL in upper limb. Mean age = 60 ± 7 (range 54–78)	Group 1: Flexitouch (Portable device to simulate the effects of manual lymphatic drainage), followed by standard treatment (compression and daily self-administered massage) (n = 5) Group 2: Standard treatment, followed by Flexitouch (n = 5).	36-Item Short Form Health Survey (SF-36).	There were no significant differences in HRQoL between Flexitouch and standard treatment.	9
Williams AF, Vadgama A, Franks PJ, Mortimer PS (2002) [21] Midlothian, Stotland.	Randomized, controlled, and crossover study of two groups.	31 women with BCRL in upper limb. Mean age group 1 = 59.7 ± 2.1 Mean age group 2 = 59.3 ± 2.4	Group 1: Manual lymphatic drainage, followed by simple lymphatic drainage (n = 15). Group 2: Simple lymphatic drainage, followed by manual lymphatic drainage (n = 16).	EORTC Core Quality of Life questionnaire (EORTC QLQ-C30).	Manual lymphatic drainage improved emotional function in terms of reduced worry, irritability, tension, and feelings of depression (*p* = 0.006). It also improved dyspnea (*p* = 0.04) and reduced sleep disturbances (*p* = 0.03). The other subscales did not reach statistical significance. Simple lymphatic drainage did not result in significant changes in any of the HRQoL parameters.	7
Tidhar D, Katz-Leurer M (2010) [22] Ramat Gan, Israel.	Randomized case-control study of two groups.	48 women with mild, moderate, and severe BCRL in upper limb. Mean age study group = 56.2 ± 10.7 Mean age control group = 56.5 ± 8.8	Study group: Aquatic lymphatic therapy and self-management measures (n = 16). Control group: Only self-management measures (n = 32).	Upper Limb Lymphedema-27 (ULL-27).	Significant differences were found between the groups after the intervention, in the emotional dimension (*p* = 0.03) and in the social dimension (*p* = 0.01). An improvement was observed in the HRQoL of the patients in the study group and a decrease in the patients in the control group.	9
Haghighat S, Omidi Z (2021) [23] Teheran, Iran.	Randomized clinical trial of two groups.	70 women with BCRL in upper limb. Mean age = 51.42 ± 9.71	Group 1: Routine lymphedema treatment and virtual self-care education program (n = 35). Group 2: Routine lymphedema treatment and in-person self-care education program (n = 35).	EORTC Core Quality of Life questionnaire (EORTC QLQ-C30). EORTC Core Quality of Life questionnaire (EORTC QLQ-BR23).	After finishing the intervention, the HRQoL of the in-person group improved compared to the virtual group, in physical function (*p* = 0.006), roles (*p* = 0.026), emotional (*p* = 0.014), and social (*p* = 0.023). Three months after the intervention, scores in global health (*p* = 0.21), physical function (*p* = 0.004), roles (*p* = 0.009), emotional (*p* < 0.001), and social (*p* = 0.048) were better in in-person group.	7
Tsai H-J, Hung H-C, Yang J-L, Huang C-S, Tsauo J-Y (2009) [24] Taipei, Taiwan.	Randomized and controlled study of two groups.	42 women with mild, moderate, and severe BCRL in upper limb. Mean age = 54.6 (range 36–75)	Group 1: Decongestant lymphatic treatment group (bandage) and pneumatic compression (n = 21). Group 2: Modified decongestant lymphatic treatment group (K-tape) and pneumatic compression (n = 21).	EORTC Core Quality of Life questionnaire (EORTC QLQ-C30) EORTC Core Quality of Life questionnaire (EORTC QLQ-BR23).	Post-intervention, there was a statistically significant improvement in role-playing quality of life in the K-tape group. After the follow-up three months after the end of the intervention, no significant differences were found between the two groups, except in emotional function, where it improved in the bandage group and deteriorated in the K-tape group (*p* < 0, 05).	9
Basha MA, Aboelnour NH, Alsharidah AS, Kamel FH (2022) [25] El Cairo, Egypt.	Randomized and controlled study of two groups.	60 women older than 30 years, with stage I, II or III of BCRL in upper limb. Mean age group 1 = 48.83 ± 7.00 Mean age group 2 = 52.07 ± 7.48	Group 1: Xbox group received VR Kinect-based games (n = 30). Group 2: Resistance exercise group received resistance training (n = 30). Both groups received complex decongestive physiotherapy (manual lymphatic drainage, compression bandages, skin care and exercises).	36-Item Short Form Health Survey (SF-36).	After the intervention, the group that used the Xbox Kinect improved some domains with respect to the group that performed resistance exercises, in bodily pain (*p* = 0.002), general health (*p* < 0.001) and vitality (*p* = 0.006).	9
Omidi Z, Kheirkhah M, Abolghasemi J, Haghighat S (2020) [26] Teheran, Iran.	Randomized clinical trial of three groups.	70 women with stage I, II or III of BCRL in upper limb. Mean age group 1 = 52.47 ± 10.62 Mean age group 2 = 50.44 ± 8.81 Mean age group 3 = 50.23 ± 8.90	Group 1: Group-based education and routine treatments (n = 35). Group 2: Education based on social networks and routine treatments (n = 35). Group 3: Control group only with routine treatments (n = 35).	Lymphedema Life Impact Scale (LLIS).	After the intervention, statistically significant differences were found between the groups. Group education was more effective than social network-based training in improving psychosocial domain (*p* < 0.05).	9
Belmonte R, Tejero M, Ferrer M, Muniesa JM, Duarte E, Cunillera O, Escalada F (2012) [27] Spain.	Randomized, controlled, and crossover study of two groups.	36 women with BCRL in upper limb. Mean age = 67.78 ± 11.30	Group 1: manual lymphatic drainage sessions, followed by low intensity and low frequency electrotherapy sessions (n = 19). Group 2: low intensity and low frequency electrotherapy sessions, followed by manual lymphatic drainage (n = 17).	Functional Assessment of Cancer Therapy Questionnaire for Breast Cancer version (FACT-B + 4).	FACT-General, FACT-Breast and trial outcome index summaries increased significantly between pre and post low-frequency low-intensity electrotherapy treatment evaluations (*p* < 0.05). No statistically significant changes were found between the evaluations before and after treatment with manual lymphatic drainage.	9
Han K, Kwon O, Park H-J, Kim A-R, Lee B, Kim M, Kim J-H, Yang C-S, Yoo H-S (2020) [28] Daejeon, South Korea.	Single-arm pilot clinical trial.	10 women older than 19 years, with BCRL in upper limb. Mean age = 53.0 (range 45–60)	The scores prior to the intervention are considered control measures. Subsequently all patients are treated with electronic Moxibustion.	EORTC Core Quality of Life questionnaire (EORTC QLQ-BR23).	No significant differences were found throughout the intervention.	5
Muñoz-Alcaraz MN, Pérula-de Torres LA, Jiménez-Vílchez AJ, Rodríguez-Fernández P, Olmo-Carmona MV, Muñoz-García MT, Jorge-Gutiérrez P, Serrano-Merino J, Romero-Rodríguez E, Rodríguez-Elena L, et al. (2022) [29] Cordoba and Aragon, Spain.	Prospective, randomized, and case-control clinical trial of two groups.	63 women with stage I or II of BCRL in upper limb. Mean age = 59.24 ± 9.55	Study group: Treatment based on activity-oriented antiedema proprioceptive therapy (without compression on the affected upper limb and using activity as treatment method) (n = 32). Control group: Usual guidelines (preventive measures, skin care, exercise, prescription of compression garments, multilayer bandages, and manual lymphatic drainage) (n = 31).	Upper Limb Lymphedema 27 Value (ULL-27).	Immediately after the end of therapy, the group of participants who underwent treatment based on activity-oriented antiedema proprioceptive therapy improved their HRQoL in the social dimension compared to the group that received conventional treatment (*p* < 0.05). No statistically significant differences were obtained in the rest of the HRQoL dimensions analyzed, nor in the measurement taken three months after the end of the treatment.	9
Cormie P, Pumpa K, Galvão DA, Turner E, Spry N, Saunders C, Zissiadis Y, Newton RU (2013) [30] Perth and Canberra, Australia.	Prospective, randomized clinical trial of three groups.	62 women with stage I, II or III of BCRL in upper limb. Mean age group 1 = 56.1 ± 8.1 Mean age group 2 = 57.0 ± 10.0 Mean age group 3 = 58.6 ± 6.7	Group 1: Group sessions with high load resistance exercises (n = 22). Group 2: Group sessions with low-load resistance exercises (n = 21). Group 3: Control group. Usual care control (n = 19).	36-Item Short Form Health Survey (SF-36).	After the intervention, changes in the physical functioning domain were found that were significantly greater in the high and low load groups compared to the control group (*p* = 0.040). No significant differences were observed between groups in any of the other HRQoL domains tested.	7
De Vrieze T, Gebruers N, Nevelsteen I, Fieuws S, Thomis S, De Groef A, Tjalma WAA, Belgrado J-P, Vandermeeren L, Monten C, et al. (2022) [31] Belgium.	Randomized, controlled, and multicenter clinical trial of three arms.	194 women with stage I or II of BCRL in upper limb. Mean age group 1 = 60.0 ± 11.0 Mean age group 2 = 62.0 ± 10.0 Mean age group 3 = 61.0 ± 9.0	Group 1: Decongestive lymphatic therapy, with fluoroscopy-guided manual lymphatic drainage (n = 65). Group 2: Decongestive lymphatic therapy, with traditional manual lymphatic drainage (n = 64). Group 3: Decongestive lymphatic therapy, with placebo manual lymphatic drainage (n = 65).	McGill-QoL Questionnaire.	Differences in HRQoL between groups were small (< 0.5 McGill-QoL total score points), showing narrow confidence intervals, indicating negligible differences in effect between the three interventions.	10
McNeely ML, Dolgoy ND, Rafn BS, Ghosh S, Ospina PA, Al Onazi MM, Radke L, Shular M, Kuusk U, Webster M, et al. (2022) [32] Edmonton, Calgary and Vancouver, Canada.	Randomized, controlled, and multicenter clinical trial of three arms.	120 women with BCRL in upper limb. Mean age = 61.0 ± 11.0	Group 1: Daytime compression garment alone (standard care) (n = 39). Group 2: Daytime compression garment plus nighttime compression bandaging (n = 44). Group 3: Daytime compression garment plus the use of a nighttime compression system garment (n = 37).	Lymphedema Functioning, Disability, and Health Questionnaire (Lymph-ICF).	Statistically significant (*p* < 0.05) within-group changes for HRQoL were observed in all groups as measured by Lymph-ICF, implying benefits across all three interventions. No significant differences were found between the three interventions.	9
Dayes IS, Whelan TJ, Julian JA, Parpia S, Pritchard KI, D’Souza DP, Kligman L, Reise D, LeBlanc L, McNeely ML, et al. (2013) [33] Canada.	Randomized and case-control clinical trial of two groups.	103 women with BCRL in upper limb. Median age study group = 61.0 ± 10.7 (range 36–86) Median age control group = 59.0 ± 8.8 (range 41–76)	Study group: daily manual lymphatic drainage and bandaging followed by compression garments (n = 57). Control group: Compression therapy only (elastic compression garments) (n = 46).	36-Item Short Form Health Survey (SF-36).	No differences were found in the mean scores for the physical and mental components of the SF-36 at baseline or at any of the follow-up periods, between and within treatment groups.	7
Kim Y, Park EY, Lee H (2023) [34] Incheon, South Korea.	Randomized, cross-sectional, crossover study.	30 women with stage I or II of BCRL in upper limb. Mean age group 1 = 47.8 ± 5.2 Mean age group 2 = 48.0 ± 8.3	Group 1: myofascial release plus complex decongestant therapy, followed by a washout period, and then placebo myofascial release plus complex decongestant therapy (n = 15). Group 2: Placebo myofascial release plus complex decongestant therapy, followed by a washout period, and then myofascial release plus complex decongestant therapy (n = 15).	Functional Assessment of Cancer Therapy–Breast (FACT-B).	Significant improvements in FACT-B scores were found before and after myofascial release (*p* < 0.05). No significant differences were found before and after placebo myofascial release (*p* > 0.05).	8
Gordon LG, Battistutta D, Scuffham P, Tweeddale M, Newman B. (2005) [35] Brisbane, Australia.	Randomized and controlled clinical trial of three arms.	275 women older than 25 years with BCRL in upper limb. Mean age group 1 = 59.0 ± 10.7 Mean age group 2 = 54.0 ± 11.3 Mean age group 3 = 55.0 ± 10.3	Group 1: Domiciliary Allied Health and Acute Care Rehabilitation Team (DAART) program (n = 36). Group 2: Strength Through Recreation Exercise Togetherness Care Health (STRETCH) program (n = 31). Group 3: Control group (n = 208).	Functional Assessment of Cancer Therapy–Breast (FACT-B).	Mean HRQoL characteristics gradually improved (*p* < 0.05) in all groups from 6 to 12 months after diagnosis, and no notable differences were found.	6
Loudon, A., Barnett, T., Piller, N., Immink, M. A., & Williams, A. D (2014) [36] Hobart and Launceston, Australia.	Randomized, controlled, multicenter, and case-control clinical pilot trial of two groups.	23 women with BCRL in upper limb. Mean age = 57.6 ± 10.5 (range 34–80)	Study group: Participation in yoga classes (breathing practices, physical postures, meditation, relaxation techniques and optional compression measures (n = 12). Control group: Regular self-care (use of compression sleeves, self-massage, skin protection, and regular continued lymphatic treatment) (n = 11).	Lymphedema quality of life questionnaire (LYMQOL).	At the end of the intervention at eight months, there was a significant improvement in the intervention group compared to the control group on the quality of life symptom subscale (*p* = 0.038). In the long-term evaluation at twelve months, no significant changes were found between the groups compared to the eight-month evaluation.	7

## Data Availability

Not applicable.

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
