# Peer review of "Effect of Conservative Rehabilitation Interventions on Health-Related Quality of Life in Women with Upper Limb Lymphedema Secondary to Breast Cancer: A Systematic Review"

_healthcare, 2023, doi:10.3390/healthcare11182568_

Round 1
Reviewer 1 Report
I commend the authors for their exploration of the health-related quality of life associated with various conservative interventions in the management of breast cancer-related lymphedema in the upper limb among women. This systematic review is excellently composed, encompassing all relevant studies and effectively summarizing them in the table.
Minor Comments
- Please update the 2022 or 2023 statistics in the introduction.
- It would be better to add the clinical trial phase also in the table.
Author Response
Please see the attachment file and v.2 of the manuscript.

Reviewer 2 Report
The authors undertook a systemic review of studies examining the impact of conservative therapies on the health-related quality of life (HRQoL) affected by breast cancer-related lymphedema (BCRL) in women undergoing breast cancer treatment. The manuscript is well written and can be an important addition to the knowledge base of HRQoL with some revision. Studies included in the review use a heterogeneous set of therapies to examine the variable HRQoL; in addition, there is heterogeneity in the method of assessment of this variable. It is important for the review to address the inherent heterogeneity that exists when examining this variable with respect to therapies administered and how their impact is measured. Finally, when presenting the results, it is necessary to include both class intervals as well as the p-value of each study where the results were significant. Again, since the nature of the variable is such that there will be heterogeneity in the outcome, it will be helpful if the authors could stratify the outcomes based on a category such as impact on physical or emotional symptoms. Such stratification can also be applied to treatments. In Table 3, it would be helpful if the authors could split the interventions into two: one showing the comparisons for the specific study and the other the treatment administered.
The conclusion of the study needs to address the findings of the review and recommendations for further studies and reviews. Perhaps studies examining the impact of more uniform measures of treatment with a defined aspect of HRQoL within a similar age range while addressing heterogeneity in socio-economic status could be helpful in deciphering the impact of conservative therapies on HRQoL in women treated for breast cancer-related lymphedema while undergoing breast cancer treatment.
The manuscript has some minor typological errors that should be addressed before publication.
The manuscript has some minor typological errors that should be addressed before publication.
Author Response

(The authors gave the same response as above.)

Reviewer 3 Report
This is important topic although it is not innovative in the current research field. Valuable to be published.
Conclusion:
even though results are controversial, please re-summarize what were the most effect treatment regimen to improve HRQOL and suggest future research.
overall English editing is required.
Author Response

(The authors gave the same response as above.)

Reviewer 4 Report
Dear authors,
Thank you very much for a well-written and interesting review. The topic is important as the frequency of lymph oedema is high in the target group, and there are few interventions in the field. HRQoL is an essential measure regarding interventions in this group.
I have some comments:
Line 70-75: Search strategy. From what I understand you have combined all Mesh terms in all databases. Usually, one starts with fewer terms and narrows it down by combining. I cannot see that you have used intervention. Why not?
Line 76-81. Eligibility criteria. I lack a limitation of the publication year. You have a span of 20 years. Limiting the number of years when performing a systematic review is evident. It clearly will affect the results and needs to be discussed in the limitation of the study section, which is missing in your manuscript.
Check the spelling in Figure 1.
Line 100-108: Results: The article selection must be moved to the methods before quality assessment. It is not results.
Line 109-127 Quality assessment: The paragraph is a bit difficult to follow and should benefit from a thorough rephrasing. As it is written now, the reader needs to look at the footnote under the table to be able to follow.
Results:
Start with describing the sample in the included studies, the number of participants the years where they originate from etc. After that, you present the results of the review. As you are not performing a meta-analysis this part of the results needs to be elaborated a lot. Table 3 is only for an overview not to replace your results. I can see that you have written information about your studies in the discussion section that would suit better in the results. Please, rewrite.
Table 3: It is well presented and clear. However, I suggest adding a column to the right that indicates the quality assessment scale value. It would certainly facilitate for the reader.
Discussion:
You do not discuss your results with the findings of other studies. Furthermore, a discussion about your methodology with limitations is missing. As stated above, the results and the discussion need to be elaborated.
Thank you!
Author Response

(The authors gave the same response as above.)
